EMBO
Molecular Medicine

# CXCL12α/SDF-1 from perisynaptic Schwann cells promotes regeneration of injured motor axon terminals

Samuele Negro[1], Francesca Lessi[2], Elisa Duregotti[1], Paolo Aretini[2], Marco La Ferla[2], Sara Franceschi[2], Michele Menicagli[2], Elisanna Bergamin[1], Egle Radice[3], Marcus Thelen[3], Aram Megighian[1], Marco Pirazzini[1], Chiara M Mazzanti[2],* (iD), Michela Rigoni[1],** (iD) & Cesare Montecucco[1,4],*** (iD)

## Abstract

The neuromuscular junction has retained through evolution the capacity to regenerate after damage, but little is known on the inter-cellular signals involved in its functional recovery from trauma, autoimmune attacks, or neurotoxins. We report here that CXCL12α, also abbreviated as stromal-derived factor-1 (SDF-1), is produced specifically by perisynaptic Schwann cells following motor axon terminal degeneration induced by α-latrotoxin. CXCL12α acts via binding to the neuronal CXCR4 receptor. A CXCL12α-neutralizing antibody or a specific CXCR4 inhibitor strongly delays recovery from motor neuron degeneration *in vivo*. Recombinant CXCL12α *in vivo* accelerates neurotransmission rescue upon damage and very effectively stimulates the axon growth of spinal cord motor neurons *in vitro*. These findings indicate that the CXCL12α-CXCR4 axis plays an important role in the regeneration of the neuromuscular junction after motor axon injury. The present results have important implications in the effort to find therapeutics and protocols to improve recovery of function after different forms of motor axon terminal damage.

**Keywords** CXCL12; CXCR4; neuromuscular junction; neuroregeneration; perisynaptic Schwann cells

**Subject Categories** Neuroscience; Regenerative Medicine

## Introduction

The neuromuscular junction (NMJ) is a specialized tripartite synapse formed by the motor axon terminal (MAT) and the muscle fiber (MF), which are separated by a basal lamina (BL), and covered by perisynaptic Schwann cells (PSCs). This is the best-studied synapse, and most of what is presently known about development, structure, and function of a synapse derives from studies on the NMJ. The NMJ is not protected by anatomical barriers, and it is subjected to traumas and to the attack of many pathogens, including neurotoxins and autoimmune antibodies. For these reasons and for its essential role in life and survival, the NMJ, unlike the central nervous system synapses, has retained throughout vertebrate evolution the capacity to regenerate (Brosius Lutz & Barres, 2014). Damaged MATs regenerate readily forming new NMJs that look and perform as the original ones (Son *et al*, 1996; Sanes & Lichtman, 1999; Darabid *et al*, 2014). Hence, the NMJ is a privileged system to study the inter- and intracellular signaling occurring during MAT degeneration and, more importantly, of those governing the ensuing regeneration.

We have recently established a murine model of NMJ degeneration and regeneration based on the specific and acute actions of α-latrotoxin (α-LTx) or of snake presynaptic PLA$_2$ neurotoxins (Dixon & Harris, 1999; Rigoni *et al*, 2004, 2005; Ushkaryov *et al*, 2008), with the aim of identifying the molecules involved in intra- and inter-cellular signaling governing NMJ regeneration (Duregotti *et al*, 2015; Rigoni & Montecucco, 2017). These neurotoxins bind specifically to the presynaptic membrane and induce a large Ca$^{2+}$ entry into nerve terminals, with exocytosis of synaptic vesicles, rounding and degeneration of mitochondria, and Ca$^{2+}$-induced activation of hydrolases such as calpains (Rigoni *et al*, 2007, 2008; Tedesco *et al*, 2009; Duregotti *et al*, 2013). This causes the complete MAT degeneration with formation of debris that are phagocytosed by PSCs, without recruitment of inflammatory cells. PSCs become activated during MAT degeneration by signals which include mitochondrial and cytosolic *alarmins* (Duregotti *et al*, 2015; Negro *et al*, 2016). PSCs play an essential role in regeneration, not

1   Department of Biomedical Sciences, University of Padua, Padua, Italy
2   Laboratory of Genomics, Pisa Science Foundation, Pisa, Italy
3   Institute for Research in Biomedicine, Università della Svizzera Italiana, Bellinzona, Switzerland
4   CNR Institute of Neuroscience, Padua, Italy
    *Corresponding author. Tel: +39 050974061; E-mail: c.mazzanti@fondazionepisascienza.org
    **Corresponding author. Tel: +39 0498276077; E-mail: michela.rigoni@unipd.it
    ***Corresponding author. Tel: +39 0498276058; E-mail: cesare.montecucco@unipd.it

limited to the clearing of necrotic debris (Son *et al*, 1996; Sanes & Lichtman, 1999; Jessen *et al*, 2015). Stimulated PSCs, together with MF, are believed to produce and release a series of factors that act on the motor axon stump inducing its growth until contacts with the BL are re-established, with resumption of regulated neurotransmitter release (Son *et al*, 1996; Sanes & Lichtman, 1999; Darabid *et al*, 2014).

In mice, the entire degeneration and regeneration process induced by α-LTx is completed within about 5 days with no apparent inflammatory response, thus making the task of identifying inter-cellular signals simpler. We carried out a transcriptome analysis of the NMJ at different time points after injection of α-LTx, to identify novel intra- and inter-cellular signaling molecules. Among the many messenger RNAs (mRNAs) differentially expressed during degeneration and regeneration, we investigated the role of the chemokine CXCL12α (also abbreviated as stromal-derived factor 1, SDF-1). Here, we provide compelling evidence that CXCL12α is produced by activated PSCs and plays an important role in NMJ regeneration by inducing the growth of the motor axon.

# Results

## CXCL12α mRNA increases during NMJ degeneration

To determine the transcriptome program of murine NMJs during acute degeneration of MAT followed by complete regeneration, we used an animal model that we recently developed (Duregotti *et al*, 2015). The time course of α-LTx-induced reversible degeneration was defined by staining over time the presynaptic markers SNAP25 (synaptosomal-associated protein 25) and VAMP1/synaptobrevin. We employed transgenic mice expressing a cytosolic GFP in Schwann cells (SCs) under the *plp* promoter (Mallon *et al*, 2002) to visualize PSCs. In *Levator auris longus* (LAL) muscles (Angaut-Petit *et al*, 1987) injected with α-LTx, there is a progressive fragmentation of presynaptic nerve terminals that peaks at 4 h; with time, a gradual reappearance of SNAP25 (Fig 1A) and VAMP1 (Fig 1B) takes place, with the motor axon end plate returning similar to controls by 96 h. Therefore, we performed a transcriptome analysis of NMJs from LAL, a muscle very suitable for NMJs collection by laser microdissection, from samples taken at 4, 24, 96 h after α-LTx injection. Total RNA was extracted, quality-controlled, retro-transcribed, amplified, and sequenced (Fig EV1), and results were compared with those of controls.

Among the mRNAs differentially expressed during MAT degeneration and regeneration, we focused on the one encoding for CXCL12α, which is poorly expressed in controls, upregulated at 4 h (MAT degenerated and active phagocytosis of debris by PSCs), and returns to low expression later on (Fig 1C). Such pattern of CXCL12α mRNA change with time was confirmed also in soleus muscle by droplet digital PCR (Figs 1D and EV2).

CXCL12α was previously reported to play an active role in neuronal development (Lieberam *et al*, 2005; Arnò *et al*, 2014; Shellard & Mayor, 2016). Given that development and regeneration may share molecules and signaling pathways, we tested the possibility that CXCL12α plays a role in the regrowth of the motor axon stump to reform a functional NMJ.

## CXCL12α is expressed by PSCs during NMJ degeneration

The sampling protocol developed here provides RNA from MAT, PSCs, and MF at the same time; therefore, RNA transcripts cannot be directly attributed to one cell type. To identify the cellular origin of CXCL12α, we performed at the NMJ both fluorescence *in situ* hybridization (FISH) and immunohistochemistry. CXCL12α mRNA signal is barely detectable in control NMJs, but becomes evident 4 h post-injection within PSCs (Fig 1E). Using a specific antibody, we found that CXCL12α is contained inside PSCs granules (56 ± 5% of CXCL12α-positive NMJs; Fig 1F, *arrows* and orthogonal projection). In agreement with the transcriptomic profile, the intensity of CXCL12α staining decreases with time, becoming practically absent at 96 h, when regeneration is well under way. These data indicate that the production of this chemokine by PSCs is an early event.

## Neutralization of CXCL12α delays NMJ regeneration *in vivo*

CXCL12 knockout mice are not viable (Nagasawa *et al*, 1996), and in conditional CXCL12α knockout mice protein disappearance is incomplete (Arnò *et al*, 2014). To assess whether CXCL12α plays a role in the regeneration of damaged NMJs, we followed the classical approach used for nerve growth factor (Angeletti *et al*, 1971). We injected intraperitoneally a CXCL12α-neutralizing IgG monoclonal antibody before α-LTx injection in the mice hind limb. By this approach, the biochemical knockout of the chemokine begins rapidly after antibody injection and lasts for a long time, as the half-life of murine IgG antibodies exceeds 11 days (Sigounas *et al*, 1994). Recovery of muscle contraction was measured by recording the evoked junctional potentials: NMJ regain of function is significantly delayed at 72 and 96 h in the presence of the anti-chemokine antibody (Fig 2A), in line with its slower anatomical recovery, as assessed by immunostaining the presynaptic marker VAMP1 (Fig EV3). Injection of the sole anti-chemokine antibodies has no effect on neurotransmission. As expected, the impairment of regeneration is not complete, given that the process is driven and governed by a set of factors (Sanes & Lichtman, 1999; Darabid *et al*, 2014). In this light, the effect of CXCL12α on nerve recovery of function is even more remarkable.

## CXCL12α promotes axonal growth in cultured spinal cord motor neurons

In cultured cerebellar and hippocampal neurons, CXCL12α was shown to promote axonal elongation (Arakawa *et al*, 2003; Pujol *et al*, 2005). Accordingly, we tested the effect of recombinant CXCL12α (*r*CXCL12α) on primary cultures of spinal cord motor neurons (SCMNs) grown in culture dishes (Arce *et al*, 1999). Figure 2B shows that *r*CXCL12α (500 ng/ml) strongly stimulates axon elongation, and panel C provides a quantitative estimation of this effect. This concentration was used before (Opatz *et al*, 2009; Heskamp *et al*, 2013), and it appears appropriate if one considers that the chemokine is released from PSCs into the very small volume between nerve terminals and PSCs. The growth-promoting effect of CXCL12α can be better appreciated in microfluidic devices, which consist of a somatic chamber, where neurons are plated, connected to a distal chamber, containing the chemokine,

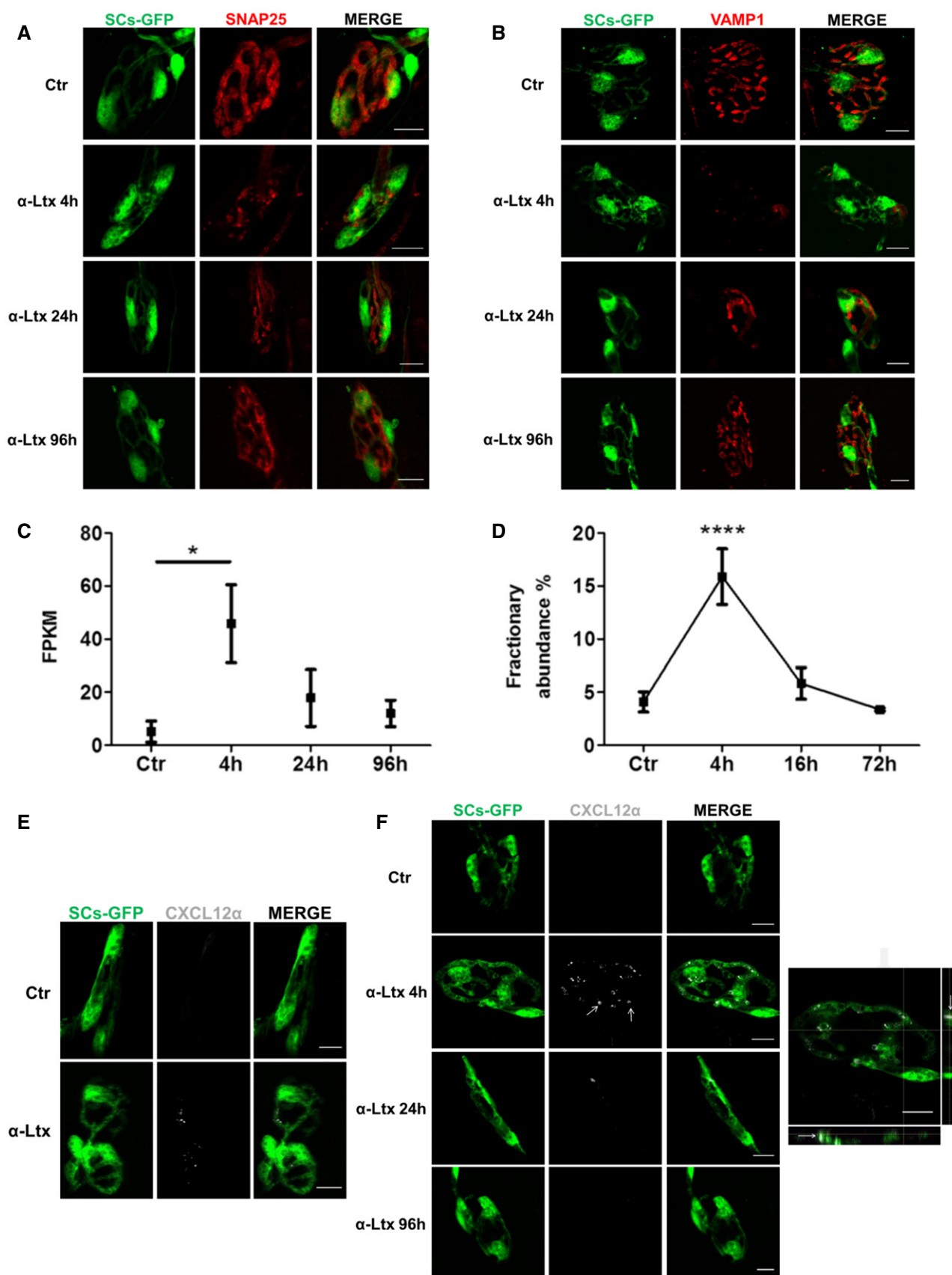

**Figure 1.**

◄

**Figure 1.  CXCL12α is expressed by perisynaptic Schwann cells during nerve terminal degeneration.**

A, B   The time course of motor axon terminal degeneration and regeneration induced by α-LTx at LAL NMJs was determined in mice with GFP-expressing SCs (green), using the presynaptic markers SNAP25 (red) (A) and VAMP1 (red) (B); at 4 h, SNAP25 and VAMP1 are phagocytosed inside PSCs. Muscles were fixed 0, 4, 24, and 96 h post-injection. Scale bars: 10 μm.

C   CXCL12α mRNA levels (expressed as FPKM, fragments per kilobase of exon per million fragments mapped) during MAT degeneration and regeneration induced by α-LTx at LAL NMJs. Data are presented as mean ± SD. *P = 0.017 (ctr vs. 4 h), three independent experiments. Statistical analysis was performed by Cuffdiff software.

D   CXCL12α mRNA levels measured by droplet digital PCR performed on cDNA from soleus muscles locally injected with α-LTx (0, 4, 16, and 72 h). The time points analyzed were chosen on the basis of the time course of nerve terminal degeneration and regeneration by α-LTx poisoning in soleus muscle, which is slightly different from that of LAL. Nonetheless, in both muscles degeneration peaks at 4 h (Fig EV2). Data are expressed as fractionary abundance with respect to the housekeeping GAPDH, five independent experiments. Data are presented as mean ± SD. ****P < 0.0001 (4 h vs. ctr, 16 and 72 h) by ANOVA with *post hoc* Tukey test.

E   *In situ* CXCL12α mRNA hybridization (white) at soleus NMJ before and after 4-h intoxication with α-LTx. PSCs are in green. Representative images are shown. Scale bars: 10 μm.

F   Immunostaining for CXCL12α (white, arrows) at LAL NMJs in controls and after 4, 24, and 96 h of intoxication. PSCs are in green. Scale bars: 10 μm. Right: Orthogonal projection of α-LTx-poisoned NMJ (4 h) shows that CXCL12α spots are inside PSCs (arrows). Scale bar: 10 μm.

via parallel micrometer-size grooves, within which the axons grow (Park *et al*, 2006; Zahavi *et al*, 2015; Fig 2D–F). In these microfluidic plates, the stimulus is applied directionally as a gradient from the distal chamber versus the axon tip. An additional advantage of these devices is that one can measure number and length of single axons growing within the groove. *r*CXCL12α failed to promote axonal elongation when applied to the somatic chamber, suggesting that it acts by interacting with axonal receptors (Fig EV4).

Remarkably, local administrations of *r*CXCL12α after neurotoxin injection in the mice soleus muscle accelerated the functional recovery of degenerated nerve terminals, as assessed by electrophysiological recordings (Fig 2G).

### The CXCL12α-CXCR4 axis directs motor axon growth *in vivo* and *in vitro*

In the immune and nervous systems, CXCL12α signals mainly via the CXCR4 receptor (Chalasani *et al*, 2003; Lieberam *et al*, 2005; Pujol *et al*, 2005; Guyon, 2012; Nagasawa, 2014). This receptor is well expressed in cultured SCMNs: it concentrates at the growing tips of axons (Fig 3A, arrow), similar to what found in isolated brain neurons *in vitro* (Arakawa *et al*, 2003; Pujol *et al*, 2005), and co-localizes with the typical growth cone-associated marker GAP43 (Fig 3B).

In the adult NMJs, CXCR4 signal is almost undetectable, but becomes evident in the motor axon stump upon α-LTx-induced degeneration (Fig 3C), reminiscent of the transient expression of CXCR4 in developing spinal cord neurons (Lieberam *et al*, 2005). No CXCR4 was detected by antibody staining in PSCs, nor it was revealed by our NMJ transcriptome analysis, which does not include mRNAs from neuronal cell bodies.

AMD3100 is a specific CXCR4 antagonist (De Clercq *et al*, 1992; Donzella *et al*, 1998). When AMD3100 was applied to the somatic chamber of microfluidic devices, the promotion of axon growth exerted by *r*CXCL12α was reduced (Fig 3D and E). The same result was obtained with SCMNs grown in culture dishes (Fig EV5).

AMD3100, administered intraperitoneally in mice, delays NMJ recovery from paralysis caused by a local injection of α-LTx in the hind limb, monitored by electrophysiology (Fig 3F). This result provides a pharmacological evidence of the involvement of a functional CXCL12α-CXCR4 axis in NMJ regeneration *in vivo*.

The CXCL12α-CXCR4 axis is modulated by the atypical chemokine receptor 3 (ACKR3; Cruz-Orengo *et al*, 2011; Bachelerie *et al*, 2013), which acts as a scavenger for the chemokine (Naumann *et al*, 2010; Abe *et al*, 2014). However, we did not detect ACKR3 on nerve terminals of LAL muscle in transgenic mice expressing the GFP-ACKR3 construct (Cruz-Orengo *et al*, 2011; not shown).

Altogether, these findings indicate that the CXCL12α-CXCR4 axis operates in MAT recovery of function after an acute degeneration, supporting the view that NMJ regeneration recapitulates at least some of the events underpinning its development (Sanes & Lichtman, 1999; Darabid *et al*, 2014).

## Discussion

The major findings of the present paper are as follows: (i) CXCL12α is produced at the adult NMJ by PSCs activated by signals from injured motor axon terminals; (ii) this chemokine plays a relevant role in the process of NMJ regeneration *in vivo* by interacting with the receptor CXCR4 expressed on the motor neuron stump. These results have a double value: a translational one, as *r*CXCL12α could be included in a mixture of factors to be used to regenerate damaged NMJs, and a basic science indication that MAT regeneration recapitulates, at least in part, motor neuron development.

These results were obtained with an acute model of MAT degeneration and regeneration, which is highly reproducible and has a well-defined time course. Moreover, there appears to be little, if any, inflammation, differently from the "classical" cut and/or crush models (Rigoni & Montecucco, 2017). This model is therefore not blurred by inflammatory mediators in the search of signals exchanged among the partners of the NMJ, thus allowing one to identify immune signals possibly involved in the nervous system, as it is the case of the present work.

We have defined the time course of α-LTx-induced MAT degeneration and regeneration by immunostaining the NMJs of LAL, a very thin mice muscle composed of few fiber layers (Angaut-Petit *et al*, 1987), very suitable for imaging and collection of NMJs by laser microdissection, and by immunostaining and electrophysiology of the soleus muscle. Four time points were selected to determine the mRNA profile of mice NMJs during degeneration and subsequent recovery by transcriptomics.

We found that CXCL12α mRNA is upregulated during MAT degeneration and then returns to basal levels when regeneration is

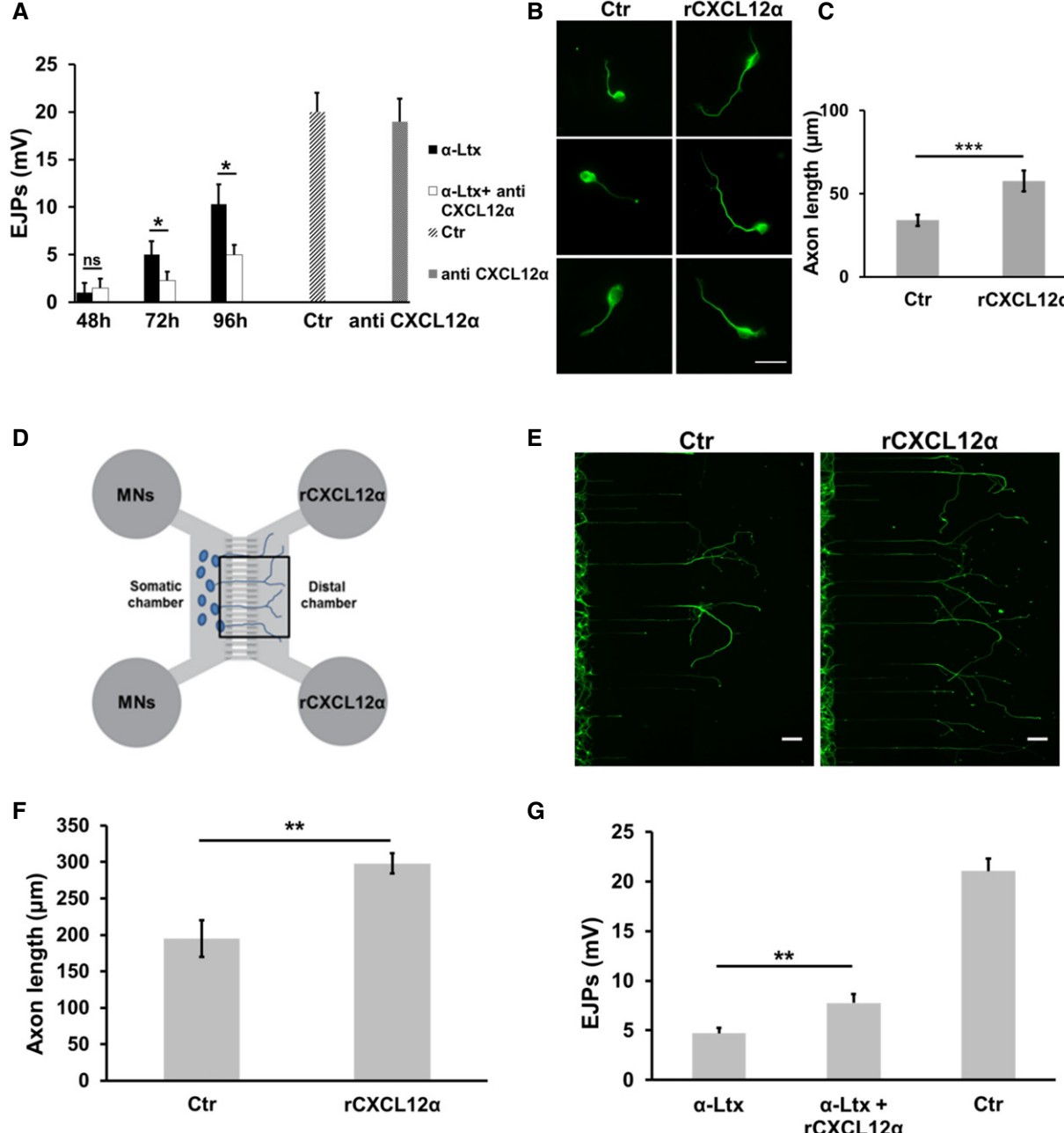

**Figure 2. The chemokine CXCL12α plays an important role in nerve terminal regeneration *in vivo*.**

A   Evoked junctional potentials (EJPs) recorded in soleus muscles 48, 72 and 96 h after α-Ltx injection in the mice hind limb, with/without a previous intraperitoneal administration of a CXCL12α-neutralizing antibody. Controlateral muscles (injected with saline) were used as controls. Muscles injected with the sole neutralizing antibody show EJPs similar to controls. Each bar represents mean ± SEM from six animals, 15 EJPs measured per animal. *P = 0.03 (72 h) and *P = 0.017 (96 h) by Student's *t*-test, unpaired, two-sided.

B, C   CXCL12α promotes axon growth of primary SCMNs. *r*CXCL12α (500 ng/ml) was added to SCMNs plated in culture dishes. After 24 h, neurons were fixed and stained for β₃-tubulin (green). Scale bar: 10 μm. Quantification is shown in (C). Each bar represents the mean ± SD from five different experiments, 70 neurons measured per experiment. ***P = 0.0003 by Student's *t*-test, unpaired, two-sided.

D   Microfluidic devices employed in the study. Somatic chambers are separated from the distal ones by a series of grooves along which SCMNs axons grow (cells plated in the somatic compartment).

E, F   *r*CXCL12α (500 ng/ml) added to the distal chamber promotes axonal elongation of SCMNs plated in the somatic chamber of microfluidic devices. The figure shows the distal chamber after 5 days of culture. Scale bars: 50 μm. Panel (F) shows the quantification of axon growth from nine experiments. Data are presented as mean ± SD. **P = 0.0019 by Student's *t*-test, unpaired, two-sided.

G   Evoked junctional potentials (EJPs) recorded in soleus muscles 72 h after α-Ltx injection in the mice hind limb, with/without local administrations of *r*CXCL12α. Controlateral muscles were used as controls. Each bar represents mean ± SEM from nine animals, 15 EJPs measured per animal. **P = 0.0043 by Student's *t*-test, unpaired, two-sided.

   

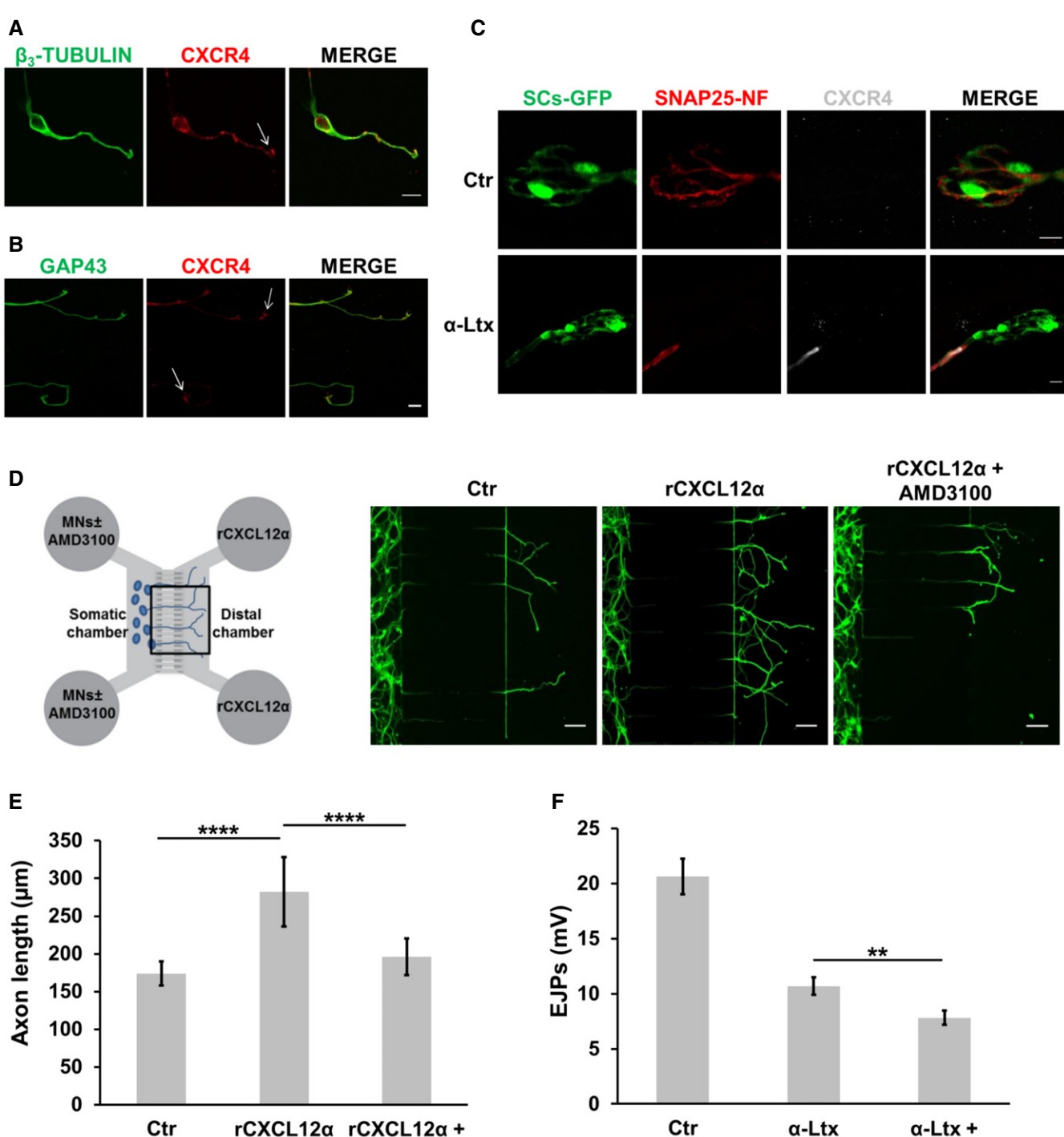

**Figure 3. The CXCL12α-CXCR4 axis participates in the regeneration of the NMJ *in vivo*.**

A, B CXCR4 (red) is expressed by primary SCMNs (β₃-tubulin positive, green) and accumulates in growing tips (arrows). Panel (B) shows CXCR4 co-localization with the growth-cone marker GAP43 (green). Scale bars: 10 μm.

C CXCR4 (white) is almost undetectable in control NMJs from LAL muscles, and it becomes evident in nerve terminals (SNAP25-NF positive, red) poisoned by α-LTx (16-h intoxication). GFP-expressing SCs are in green. Scale bars: 10 μm.

D, E AMD3100 reduces the axon elongation stimulating ability of *r*CXCL12α on SCMNs in microfluidic chambers. Axon growth was measured after 5 days of treatment. Scale bars: 10 μm. Quantification is shown in (E). Each bar represents mean ± SD from six different experiments, 70 neurons measured per experiment. ****$P < 0.0001$ by ANOVA followed by *post hoc* Tukey test.

F Intraperitoneal administration of the CXCR4 antagonist AMD3100 delays the functional recovery of mice NMJs exposed to α-LTx. Each bar represents mean ± SEM from six animals, 15 EJPs measured per animal. **$P = 0.0057$ by Student's *t*-test, unpaired, two-sided.

nearly completed. The mRNA and the encoded chemokine are specifically expressed by PSCs, arguably by the stimulation of factors released by degenerating MAT, providing a compelling molecular evidence of the active and direct involvement of PSCs in the recovery of function of the damaged NMJ. This finding supports the idea that an intense inter-cellular signaling takes place among the three partners of the NMJ not only during development (Sanes & Lichtman, 1999; Darabid *et al*, 2014), but also during regeneration in adulthood.

In the present work, the role of CXCL12α has been first tested *in vivo*, and a clear delay in regeneration was found when the chemokine was neutralized by a specific antibody. This result is remarkable if one considers that nerve regeneration is driven and assisted by an array of factors (Sanes & Lichtman, 1999; Darabid *et al*, 2014), and we neutralized only one of them by a single dose of a monoclonal antibody delivered systemically. This *in vivo* action of CXCL12α is most likely exerted via an elongation promotion effect on the motor axon, as suggested by experiments performed *in vitro* in neurons grown in microfluidic devices, where axon growth can be visualized and quantified. In addition, local administration of the recombinant chemokine accelerates NMJ functional recovery, in support of its promising therapeutic value.

CXCL12α signals via the receptor CXCR4, as suggested by the inhibitory activity *in vivo* of growth of the motor axon stump after MAT degeneration exerted by the CXCR4-specific antagonist AMD3100. This result is fully consistent with the effect of the anti-CXCL12α antibody: both agents delayed significantly the recovery of the paralyzed NMJs. Noteworthy, CXCR4 was detected by immunostaining at the tip of growing axons, and AMD3100 reduced the axon growth-stimulating activity of the chemokine. CXCR4 expression becomes detectable *in vivo* upon α-LTx-induced degeneration, supporting the involvement of this axis in axon regrowth. The absence of CXCR4 *in vivo* in PSCs excludes the possibility that CXCL12α has an autocrine effect, in agreement with the previous finding that *in vitro* the chemokine induces the death of primary SCs (Küry *et al*, 2003). In addition, the *in vitro* promotion of axonal growth by CXCL12α and the expression of CXCR4 in SCMNs indicate that primary neuronal cultures are an *in vitro* model of regeneration more than of development. As such, they appear an appropriate counterpart of the *in vivo* experiments based on antibodies and specific inhibitors reported here.

CXCL12 was discovered following a screening for proteins endowed with a secretion signal sequence (Tashiro *et al*, 1993), and soon after as a stroma-derived factor capable of stimulating the growth of the pre-B-cell population within the bone marrow, and termed therefore stromal cell-derived factor 1 (SDF-1; Nagasawa *et al*, 1994). Three variants of CXCL12 have been identified in mice: α, β, and γ. The latter isoform is resident in the nucleus, while the beta isoform is expressed by endothelial cells and appears to play a major role in leukocytes recruitment. CXCL12α is involved in the development of various regions of the central nervous system (Bagri *et al*, 2002; Chalasani *et al*, 2003; Lieberam *et al*, 2005), and the CXCL12-CXCR4 axis regulates a variety of responses in the immune and nervous systems (Lu *et al*, 2002; Stumm *et al*, 2003; Lieberam *et al*, 2005; Zhu *et al*, 2009; Guyon, 2012).

In the present work, CXCL12α was found to be produced by PSCs at the NMJs, using a combined mRNA and protein analysis. This is the first evidence for an important and defined role of CXCL12α in

the regeneration of the NMJ after MAT damage, and in the axon growth of SCMNs in culture. These results might have translational applications in the formulation of a mixture of growth factors to improve the functional recovery of the NMJ after various types of damage, including genetic neurodegenerative syndromes, and in spinal cord diseases characterized by motor neuron loss.

# Materials and Methods

### Antibodies and toxins

The following primary antibodies were employed: anti-SNAP25 (mouse monoclonal, SMI81, mouse monoclonal, Covance, catalogue 836301, 1:200), anti-VAMP1 (rabbit polyclonal, 1:200, Rossetto *et al*, 1996), anti-CXCL12α (mouse monoclonal, R&D, catalogue mab350, 1:50), anti-$\beta_3$-tubulin (rabbit polyclonal, Synaptic System, catalogue SYSY 302302, 1:200), anti-CXCR4 (rabbit polyclonal, Abcam, catalogue ab7199, 1:50–500), anti-GAP43 (rabbit monoclonal, Abcam, catalogue ab75810, 1:200), anti-neurofilaments (NF) 200 (mouse monoclonal, Sigma, catalogue N0142, 1:200), α-bungarotoxin (α-BTx, Life Technologies, catalogue B35451, 1:200). Secondary antibodies Alexa-conjugated (1:200) were from Life Technologies.

Purified α-LTx was purchased by Alomone (catalogue LSP-130). The purity of the toxin was checked by SDS–PAGE and its neurotoxicity by *ex vivo* mouse nerve-hemidiaphragm preparations as previously described (Rigoni *et al*, 2005). Unless otherwise stated, all reagents were purchased from Sigma.

### Ethical statement

C57BL/6 mice expressing cytosolic GFP under the *plp* promoter (Mallon *et al*, 2002) were kindly provided by Dr. W.B. Macklin (Aurora, Colorado) with the help of Dr. T. Misgeld (Munchen, Germany). All experimental procedures involving animals and their care were carried out in accordance with National laws and policies (D.L. n. 26, March 14, 2014) and with the guidelines established by the European Community Council Directive (2010/63/UE) and were approved by the local authority veterinary services.

### Sample preparation for laser microdissection

Upon isoflurane anesthetization, 2-month-old transgenic C57BL/6 female mice of around 20–25 g were locally injected with α-LTx close to the LAL muscles. Animals were randomized in the experimental groups based on their body weight. The toxin (5 μg/kg) was diluted in 15 μl of physiological saline (0.9% w/v NaCl in distilled water). Control animals were injected with saline. At different time points (0, 4, 24, and 96 h), treated mice were sacrificed by anesthetic overdose followed by cervical dislocation and LAL muscles were immediately fixed with a local injection of 4% PFA for 10 min. Muscles were collected, incubated with fluorescent α-BTx in sterile PBS for 30 min at 37°C, and then frozen in liquid nitrogen. Cryosections (7 μm thick) were transferred to UV-treated microscope glass slides. Microdissection was performed under direct microscopic visualization with PALM RoboMover automatic laser microdissector (Carl Zeiss, Oberkochen, Germany).

## RNA isolation

Total RNA was isolated from the microdissected samples by incubation with 50 µl of lysis buffer PKD (Qiagen, Venlo, Netherlands) and 10 µl of proteinase K solution (Promega, Madison, WI, USA) at 55°C overnight with the sample upside down. The day after samples were centrifuged for 10 min at 6,700 *g* and RNA extracted using the Maxwell® 16 LEV RNA FFPE Purification Kit (Promega) following manufacturer's instructions.

To prepare cDNA from RNA samples, the SMARTer Universal Low Input RNA kit (Clontech Laboratories) was employed following manufacturer's instructions. Libraries were prepared using the Ion TargetSeq Exome Enrichment kit, following the manufacturer's guidelines. Ion PI Sequencing 200 kit (Ion Torrent, Life Technologies). The Ion PI Chip (Ion Torrent) was prepared and calibrated for loading. The Ion PI Chip was loaded with a template-positive ISPs and run on the Ion Proton Sequencer.

## Bioinformatic analysis

Bioinformatics analysis was carried out using several command line software included in Bio-Linux (http://nebc.nerc.ac.uk/tools/bio-linux/bio-linux-7-info). Using Star aligner, the reads, previously filtered for quality and length, were processed and aligned to mouse genome (mm10 version). The unmapped reads, generated from the first step, were re-aligned by using Bowtie2. The reads mapped with Star and Bowtie2 were merged and processed with Cufflinks. Cufflinks uses the alignment file to assemble and reconstruct the transcriptome. Cuffdiff (included in Cufflinks) was used to calculate the differential gene expression between different groups (0, 4, 24, and 96 h). Cuffdiff calculates expression in two or more samples and tests the statistical significance of each observed change in expression between them. The statistical model used to evaluate changes assumes that the number of reads produced by each transcript is proportional to its abundance but fluctuates because of technical variability during library preparation and sequencing and because of biological variability between replicates of the same experiment (Trapnell *et al*, 2012). RNA-seq data have been deposited in the ArrayExpress database at EMBL-EBI (www.ebi.ac.uk/arrayexpress) under accession number E-MTAB-5730.

## Droplet digital PCR

Droplet digital PCR (ddPCR) was carried out using the ddPCRTM Supermix for Probes (No dUTP), the QX200TM Droplet Generator, the QX200 Droplet Reader, the C1000 TouchTM Thermal Cycler, and the PX1TM PCR Plate Sealer (Bio-Rad, Hercules, CA, USA) following manufacturer's instructions. Reactions were performed in triplicate in a 96-well plate using 10 µl/reaction of 2× ddPCR Supermix for Probes (No dUTP), 1 µl/reaction of 20× target primers/probe (FAM or HEX, Bio-Rad), 1 µl/reaction of 20× reference primers/probe (FAM or HEX, Bio-Rad), 3 µl cDNA and 5 µl H₂O. Detection of CXCL12 and GAPDH by ddPCR was performed using the following PrimePCR™ ddPCR™ Expression Probe Assay designed by Bio-Rad: CXCL12-FAM (ID: dMmuCPE511627, Bio-Rad) and GAPDH-HEX (ID: dMmuCPE5195283, Bio-Rad). All steps used a ramp rate of 2°C/s. Results were analyzed in the QX200 Droplet Reader, the RNA targets were quantified using the QuantaSoftTM Software (Bio-Rad), and results were expressed as fractional abundance.

## NMJ immunohistochemistry

Anesthetized mice were locally injected close to LAL muscles with α-LTx as described above. Muscles were dissected at different time points and fixed in 4% PFA in PBS for 30 min at RT. Samples were quenched, permeabilized, and saturated for 2 h in 15% goat serum, 2% BSA, 0.25% gelatine, 0.20% glycine, and 0.5% Triton X-100 in PBS. Incubation with the primary antibodies was carried out for 72 h in blocking solution. Muscles were then washed and incubated with secondary antibodies.

Images were collected with a Leica SP5 Confocal microscope equipped with a 63× HCX PL APO NA 1.4. Laser excitation line, power intensity, and emission range were chosen accordingly to each fluorophore in different samples to minimize bleed-through.

## RNA fluorescence *in situ* hybridization (FISH)

The Probe Designer tool was used to design a set of 48 oligonucleotide probes (Quasar® 570-conjugated) complementary to the *Mus musculus* CXCL12α transcript (NM_021704.3). Stellaris® RNA FISH buffers and probes were purchased by LGC Biosearch Technologies. *In situ* hybridization was performed on 7 to 10-µm-thick cryoslices according to the manufacturer's guidelines for frozen tissue. Images were collected as above.

## Electrophysiological recordings

Electrophysiological recordings were performed in oxygenated Krebs-Ringer solution on soleus muscles using intracellular glass microelectrodes (WPI, Germany) filled with one part of 3 M KCl and two parts of 3 M CH₃COOK. To determine the time course of NMJ degeneration and regeneration, anesthetized C57BL/6 female mice (2 months of age) were locally injected with the toxin in the hind limb (α-LTx 5 µg/kg), and soleus muscles were collected at 0, 4, 24, and 96 h from treatment. In another set of experiments recordings were performed on soleus muscles from mice i.p. injected with 100 µg of anti-CXCL12α antibody (diluted in 40 µl physiological solution plus 0.2% gelatine) prior to the local injection of α-LTx.

For experiments with the CXCR4 antagonist, mice were i.p. injected with 100 µg AMD3100 (in 40 µl physiological solution containing 0.2% gelatine) prior to local injection of α-LTx. AMD3100 administration was performed twice a day for 4 days.

To test the effect of chemokine administration on NMJ recovery, *r*CXCL12α was locally injected in soleus muscle (100 ng, 15 µl injection volume) 8, 24, and 48 h post-α-LTx injection. Electrophysiological recordings were performed after 72 h of intoxication.

Evoked neurotransmitter release was recorded in current-clamp mode, and resting membrane potential was adjusted with current injection to −70 mV. Evoked junction potentials (EJPs) were elicited by supramaximal nerve stimulation at 0.5 Hz using a suction microelectrode connected to a S88 stimulator (Grass, USA). To prevent muscle contraction after dissection, samples were incubated for 10 min with 1 µM µ-Conotoxin GIIIB (Alomone, Israel). Signals were amplified with intracellular bridge mode amplifier (BA-01X, NPI, Germany), sampled using a digital interface (NI PCI-6221,

National Instruments, USA) and recorded by means of electrophysiological software (WinEDR, Strathclyde University). EJP measurements were carried out with Clampfit software (Molecular Devices, USA) and statistical analysis with Prism (GraphPad Software, USA).

### Cell cultures and treatments

Primary cultures of SCMNs were prepared as described in Rigoni *et al* (2004) and exposed for 24 h to *r*CXCL12α (500 ng/ml, R&D) in culture medium. Low-density motor neuron cultures were used to test the effect of *r*CXCL12α on axon growth. In some experiments, AMD3100 (10 μM) was added to SCMNs together with *r*CXCL12α that was kept throughout the experiment. After 24 h, neurons were fixed and stained for $\beta_3$-tubulin and axon growth quantified with the ImageJ plugin NeuronJ.

### Microfluidic chambers

Microfluidic chambers were produced using established methods (Park *et al*, 2006). Polydimethylsiloxane (Dow Corning) inserts were sterilized and fixed to 50-mm glass-bottomed WillCo dishes (IntraCel) using plasma cleaning. The chambers were blocked with 0.8% BSA in PBS overnight at 37°C and then coated with poly-L-ornithine and laminin. SCMNs were plated in the somatic compartment. *r*CXCL12α (500 ng/ml) was added to the distal compartment in the absence or presence of AMD3100 (added to the somatic chamber, 10 μM). Neurons were allowed to grow for 5 days, fixed, and stained for $\beta_3$-tubulin; axon growth was quantified with the ImageJ plugin NeuronJ.

### Immunofluorescence

Cells were fixed for 15 min in 4% PFA in PBS, quenched (0.38% glycine, 0.24% $NH_4Cl$ in PBS), and permeabilized with 0.3% Triton X-100 in PBS for 5 min at room temperature. After saturation with 3% goat serum in PBS for 1 h, samples were incubated with primary antibodies (anti-$\beta_3$-tubulin, 1:200; anti-CXCR4, 1:500; anti-GAP43, 1:200), diluted in 3% goat serum in PBS overnight at 4°C, washed, and then incubated with the corresponding secondary antibodies Alexa-conjugated for 1 h at room temperature. Coverslips were mounted in Mowiol and examined by confocal (Leica SP5) or epifluorescence (Leica CTR6000) microscopy.

### Statistical analysis

Sample sizes were determined by analysis based on data collected by our laboratory in published studies. In animal studies, we used $n = 6$ mice/group for electrophysiological analysis. In cell culture studies, we performed each study with at least five independent replications. For all animal studies, we have ensured blinded conduct of experiments. For imaging analysis, the quantitation was conducted by an observer who was blind to the experimental groups. No samples or animals were excluded from the analysis.

The sample size ($N$) of each experimental group is described in the corresponding figure legend. Data displayed as histograms are expressed as means ± SEM or SD (represented as error bars). GraphPad Prism software was used for all statistical analyses. Statistical significance was evaluated using analysis of variance (ANOVA) with Tukey post-test or two-tailed, unpaired Student's *t*-test

**The paper explained**

**Problem**

The neuromuscular junction can regenerate after damage. Regeneration is driven by an intense signaling among the three main components of this synapse (motor axon terminal, perisynaptic Schwann cells, and muscle), but only few molecules governing this process have been identified so far.

**Results**

By combining a transcriptome analysis of murine NMJs during degeneration and regeneration with imaging and electrophysiological recordings, we provided strong evidence that the chemokine CXCL12α is released by a specialized type of glial cells termed perisynaptic Schwann cells and that it potently stimulates the functional recovery of the neuromuscular junction after an acute motor axon terminal degeneration. By interacting with the neuronal CXCR4 receptor, this chemokine promotes axonal elongation in primary spinal motor neurons.

**Impact**

These findings have high potential therapeutic value for the improvement of functional recovery of the neuromuscular junction after various types of insults, including genetic neurodegenerative syndromes, and in spinal cord diseases characterized by motor neuron loss.

depending on the number of groups to be analyzed. Data were considered statistically different when $*P < 0.05$, $**P < 0.01$, $***P < 0.001$, $****P < 0.0001$.

**Expanded View** for this article is available online.

### Acknowledgements

We thank W. Macklin and T. Misgeld for providing transgenic mice, and Prof. G. Martino and Dr. G. Muzio for helpful discussion. This work was supported by Fondazione CARIPARO (CM), Provincia autonoma di Trento (Bando Grandi Progetti 2012, AXonomIX) (CM), by the Interomics project of the CNR (CM), and by the Fondazione Pisana per la Scienza (CMM). MR is recipient of a Young Investigators Grant (GR-2010-2320779) from the Italian Ministry of Health.

### Author contributions

CM, MR, and CMM conceived and supervised the project. SN, MR, and CM designed and performed the experiments together with FL, ED, MLF, SF, MM, MP, MT, ER, EB, and AM, PA, CMM, SN, MR, and CM analyzed data. MR and CM wrote the manuscript with contributions and approval of all the other authors.

### Conflict of interest

The authors declare that they have no conflict of interest.

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
