## [Review Process File · EMBO Molecular Medicine]

CXCL12 α /SDF-1 from perisynaptic Schwann cells promotes regeneration of injured motor axon terminals

Samuele Negro, Francesca Lessi, Elisa Duregotti, Paolo Aretini, Marco La Ferla, Sara Franceschi, Michele Menicagli, Elisanna Bergamin, Egle Radice, Marcus Thelen, Aram Megighian, Marco Pirazzini, Chiara M Mazzanti, Michela Rigoni & Cesare Montecucco

Corresponding authors: Chiara M Mazzanti, Pisa Science Foundation
Michela Rigoni, University of Pauda
Cesare Montecucco, University of Padua

Review timeline:

Submission date:	26 October 2016
Editorial Decision:	30 November 2016
Revision received:	28 March 2017
Editorial Decision:	19 April 2017
Revision received:	03 May 2017
Accepted:	08 May 2017

Transaction Report:

Editor: Céline Carret

1st Editorial Decision

30 November 2016

Thank you for the submission of your manuscript to EMBO Molecular Medicine. I am sorry that it has taken so long to get back to you on your manuscript.

While we are still missing one referee report, given that the two evaluations we have already are consistent, and a further delay cannot be justified, I have decided to proceed based on these evaluations.

You will see that both referees are rather positive about the study and their concerns are largely overlapping. Some shortcomings are noted and suggestions proposed that if satisfactorily addressed would improve the conclusiveness of the findings and provide additional clinical relevance to the data.

Given these evaluations, I would like to give you the opportunity to revise your manuscript, with the understanding that the referees' concerns must be fully addressed and that acceptance of the manuscript would entail a second round of review. Please note that it is EMBO Molecular Medicine policy to allow only a single round of revision and that, as acceptance or rejection of the manuscript will depend on another round of review, your responses should be as complete as possible.

EMBO Molecular Medicine has a "scooping protection" policy, whereby similar findings that are published by others during review or revision are not a criterion for rejection. Should you decide to

submit a revised version, I do ask that you get in touch after three months if you have not completed it, to update us on the status.

I look forward to receiving your revised manuscript.

***** Reviewer's comments *****

Referee #1 (Comments on Novelty/Model System):

The model is appropriate and allowed to identify and then test the requirement of CXCL12/CXCR4 pathway for efficient recovery of EJPs. However, the model was not used to show sufficiency, i.e. is administration of CXCL12 accelerating recovery. If another model is more appropriate for such experiments, the authors should investigate these options. This would be important to support the medical relevance to improve recovery of function after different forms of motor axon terminal damage

Referee #1 (Remarks):

Negro et al identified perisynaptic Schwann cell-derived CXCL12 and its neuronal receptor CXCR4 as being required for motor axon regeneration after a degenerating insult, and that these findings hold potential therapeutic value. This reviewer commends the authors for their good image quality, appropriate numbers of replicates, and blinded observers/experimenters to increase confidence in the results. However, some questions remain about the significance of the work and the following revisions are suggested to increase the impact and confidence in the findings.

Major revisions

For figure 1D, qRT-PCR would be a more convincing means of determining changes in mRNA levels and offer an alternate means of validating the method that identified CXCL12 using independent biological samples. Also, the p value in the legend does not describe which time points are being compared.

Figure 1E,F and EV1 aim to convince the reader that CXCL12 is expressed in PSCs by 3D reconstruction and colocalization of CXCL12 and lysotracker. While it's reasonably convincing that that CXCL12 protein is in acidic vesicles in PSCs, it is not clear what the significance of this is. The rationale for using lysotracker was not clearly stated. One interpretation is that PSCs are endocytosing CXCL12. The authors should perform in situ hybridization (FISH) to conclude that CXCL12 is in fact expressed in PSCs. Quantification of % of NMJs expressing CXCL12 and/or # of NMJs analyzed would be appropriate.

Assuming that CXCL12 expression is in fact increased in PSCs, is it limited to PSCs or is it expressed in non-terminal Schwann cells also? qPCR on nerve would be an appropriate control.

The EFP data in figure 2 shows CXCL12 is required for behavioral recovery and the authors correlate this with in vitro data showing CXCL12 as a growth factor for motor axons. The authors should also assess anatomical recovery of NMJs with their markers in figure 1A,B in the presence and absence of CXCL12 antibody injection to determine if fewer NMJs are reoccupied as a resulting of CXCL12 inhibition.

The authors show that CXCL12/CXCR4 is necessary for efficient recovery of EJPs, but do not show sufficiency which is the main determination of therapeutic potential. An experiment testing in vivo regeneration in response to recombinant CXCL12 by intramuscular or intrathecal injection would increase the significance of the results.

Figure 3D,E aim to show that CXCR4 is expressed in neurons and not PSCs. In 3D, a counter stain with B3 tubulin to show axon tips is necessary. Also, why are there three images? In 3E, it is unclear whether CXCR4 is expressed in PSCs or axons as there is expression in domains of overlapping

green (SCs-GFP) and axons in red (Smi81, although what SMI81 stains is not defined). Maybe *in situ* hybridization is a better option to determine cell specific expression.

Minor points

Figure 1B panels labeled for Snap25 should be labeled as Vamp1.

Figure 1 sample size is not recorded in the figure legend as in other figures.

For figure 1A,B, the qualitative nature of the data is very striking, but quantification of presynaptic marker recovery would be useful to determine if variability exists and to what extent across neuromuscular junctions within the muscle.

Figure 1C may be kept but is somewhat unnecessary as an understanding of the experimental paradigm comes naturally from figure 1A,B.

In 3E, SMI81 should be Snap25 to be consistent with Figure 1.

Referee #2 (Comments on Novelty/Model System):

I believe that the adopted model is fine. It would have been great to compare the poisoning done by treating the NMJ with alpha latrotoxin with another agent (e.g. taipoxin) acting via a completely different mechanism of action

Referee #2 (Remarks):

This manuscript by Negro and colleagues describes an interesting transcriptomic approach aimed to the identification of novel signals involved in the functional recovery of the neuromuscular junction (NMJ) upon damage. The chosen insult is the treatment of the NMJ with alpha-latrotoxin, a presynaptic poison that elicits a long lasting inhibition of this specialized synaptic terminal. The authors found that stromal derived factor 1 (SDF-1 or CXCL12 alpha) is released following NMJ degeneration by schwann cells and promote regeneration via the activation of its receptor CXCR4. Sequestration of SDF-1 or inhibition of its receptor halts the functional recovery of the NMJ, suggesting that this signal transduction pathway is pivotal in NMJ regeneration.

The results are presented in a clear, succinct manner and in logical order, which makes the manuscript easy to read. In general, the claims are supported by the figures provided.

However, specific aspects of this manuscript should be revised prior to publication:

1. In contrast to the statement made in the main text and caption, both red channels in Figure 1A and B are labelled with SNAP25, rather than with SNAP25 and VAMP1. This should be amended. The specific treatment should be added to panel B. It would be useful to quantify the loss of the presynaptic markers using alpha-bungarotoxin as a mask. This analysis should address whether both plasma membrane and synaptic vesicle markers are depleted in the same extent.
2. The scheme in Figure 1C gives the wrong impression that samples are collected and/or imaged in a longitudinal rather than at fixed time points. This is misleading, especially for a scheme that should have the opposite goal. I suggest removing it.
3. No specific data emerging from the transcriptomic analysis have been provided (e.g. heat map) or if the raw data are going to be deposited in a suitable public database (e.g. Array Express at <http://www.ebi.ac.uk/arrayexpress>).
4. The staining of CXCL12 alpha is poorly visible and should be represented in white in the central panels to increase the contrast. The authors should comment on CXCL12 alpha distribution since its localization is unclear: some puncta seem to be present within the schwann cells, whereas some others map in areas not labeled in green (top in the alpha-latrotoxin 4 h sample). Likewise the statement that CXCL12 alpha puncta are colocalized with Lysotracker should be tuned down, since this colocalization is partial at best (Figure EV1). Figure EV2 seems also redundant.
5. In Figure 2A, are control samples treated with alpha-latrotoxin and either with saline or an equivalent dose of an irrelevant antibody? Please clarify the methodology in the main text.
6. Figure 2D is not very informative, since the more informative quantification of these data is provided in panel E. Does treatment with CXCL12 alpha alter neuronal survival in the conditions described in this Figure? Furthermore, is the effect on axonal elongation detected if CXCL12 alpha is only added in the somatic chamber?
7. The resolution of Figure 3D is not optimal.
8. Does CXCR4 co-localize in motoneurons with classical growth cone markers? Why is SMI81 used in Figure 3E instead of SNAP25?

Referee #1 (Comments on Novelty/Model System):

The model is appropriate and allowed to identify and then test the requirement of CXCL12/CXCR4 pathway for efficient recovery of EJPs. However, the model was not used to show sufficiency, i.e. administration of CXCL12 accelerating recovery. If another model is more appropriate for such experiments, the authors should investigate these options. This would be important to support the medical relevance to improve recovery of function after different forms of motor axon terminal damage.

Referee #1 (Remarks):

Negro et al identified perisynaptic Schwann cell-derived CXCL12 and its neuronal receptor CXCR4 as being required for motor axon regeneration after a degenerating insult, and that these findings hold potential therapeutic value. This reviewer commends the authors for their good image quality, appropriate numbers of replicates, and blinded observers/experimenters to increase confidence in the results. However, some questions remain about the significance of the work and the following revisions are suggested to increase the impact and confidence in the findings.

The experiment showing that administration of rCXCL12 α in mice accelerates NMJ recovery of function after nerve terminal degeneration is now reported in the revised paper. A positive effect obtained by simple injections of the small chemokine is indeed very remarkable, as the molecule is expected to diffuse away rather rapidly. We are currently trying to devise means of achieving a slow release at the injected site.

Major revisions

For figure 1D, qRT-PCR would be a more convincing means of determining changes in mRNA levels and offer an alternate means of validating the method that identified CXCL12 using independent biological samples.

We performed droplet digital PCR on cDNA from soleus NMJs treated with the toxin and collected by laser microdissection after 4, 16, 72 hours from toxin injection. We used soleus muscle in order to extend the validity of our transcriptome analysis performed on LAL to a different mice muscle that provides a better electrophysiological response. We compared the two muscles using immunofluorescence analysis of appropriate presynaptic markers and found that the time courses of their nerve terminal degeneration and regeneration induced by toxin injection are very similar. Digital PCR results are reported in panel D of Figure 1, and the kinetics of degeneration and regeneration in soleus muscle in Figure EV2 of the revised manuscript. Digital PCR data show the up-regulation of CXCL12 α mRNA at 4 hours (fully degenerated NMJs in both muscles) with respect to controls, similarly to what obtained previously on LAL muscle by transcriptomics.

Also, the p value in the legend does not describe which time points are being compared.

The p value of panel C refers to control vs 4 hours samples. This is now specified in Figure 1 legend of the revised manuscript.

Figure 1E,F and EV1 aim to convince the reader that CXCL12 is expressed in PSCs by 3D reconstruction and colocalization of CXCL12 and lysotracker. While it's reasonably convincing that that CXCL12 protein is in acidic vesicles in PSCs, it is not clear what the significance of this is. The rationale for using lysotracker was not clearly stated. One interpretation is that PSCs are endocytosing CXCL12.

Given the CXCL12 α -positive spots observed in PSCs at degenerating NMJs, we wished to understand the nature of such vesicles. By Lysotracker staining we showed that the chemokine is accumulated in acidic structures, whose complete characterization is the goal of a future study. Accordingly, and following the Reviewer's comment, we deleted the former Figure EV1 from the revised manuscript.

The authors should perform in situ hybridization (FISH) to conclude that CXCL12 is in fact expressed in PSCs.

Representative images of in situ hybridization at the NMJ are reported in Figure 1E of the revised manuscript, which confirm that CXCL12 α mRNA is specifically expressed in PSCs during neurodegeneration.

Quantification of % of NMJs expressing CXCL12 and/or # of NMJs analyzed would be appropriate.

The percentage of CXCL12 α -positive NMJs (56 \pm 5 %) is now reported in the Results section of the revised manuscript.

Assuming that CXCL12 expression is in fact increased in PSCs, is it limited to PSCs or is it expressed in non-terminal Schwann cells also? qPCR on nerve would be an appropriate control.

In the experimental model used here, a reproducible and controlled damage limited to the sole motor axon terminal takes place, allowing one to investigate the response of terminal SCs, whereas myelinating Schwann cells may not sense it. CXCL12 α immunostaining is barely detectable in myelinating Schwann cells, as shown by the representative images reported below.

The EFP data in figure 2 shows CXCL12 is required for behavioral recovery and the authors correlate this with in vitro data showing CXCL12 as a growth factor for motor axons. The authors should also assess anatomical recovery of NMJs with their markers in figure 1A,B in the presence and absence of CXCL12 antibody injection to determine if fewer NMJs are reoccupied as a result of CXCL12 inhibition.

We quantified the number of VAMP1-positive NMJs in soleus muscles 72 hours from toxin injection (regeneration under way), and compared it with muscles treated with both the anti-CXCL12 α antibody and the toxin. Confocal images and relative quantification are reported in Figure EV3. We found that the delay in neurotransmission recovery measured by electrophysiology due to the action of CXCL12 α neutralizing antibodies is paralleled by a reduced number of regenerating NMJs.

The authors show that CXCL12/CXCR4 is necessary for efficient recovery of EJPs, but do not show sufficiency which is the main determination of therapeutic potential. An experiment testing in vivo regeneration in response to recombinant CXCL12 by intramuscular or intrathecal injection would increase the significance of the results.

We performed EJP recordings on soleus muscles locally injected with the toxin with or without subsequent intramuscular administrations of rCXCL12 α . A faster recovery of NMJ functionality was observed in the former condition. Results are reported in panel G of Figure 2 of the revised manuscript and in the Results section. This result is remarkable as CXCL12 α is a small molecule that is expected to be washed out rapidly after injection. Therefore only those molecules that can bind their targets within a restricted time window, defined by the lymph flow rate in the muscle, will be able to exert their action at the NMJ. Moreover, it is believed that multiple factors are involved in recovery. We are currently trying to find a way of achieving a slow release of CXCL12 α after injection.

Figure 3D,E aim to show that CXCR4 is expressed in neurons and not PSCs. In 3D, a counter stain with B3 tubulin to show axon tips is necessary.

The counterstain of CXCR4 with β_3 -tubulin in cultured SCMNs is now reported in panel A of Figure 3 of the revised manuscript. At toxin-treated LAL NMJs with GFP-SCs, CXCR4 becomes expressed in the motor axon stump, where it co-localizes with neurofilaments (please see the novel confocal images reported in Figure 3C of the revised manuscript).

Also, why are there three images?

The three images simply aimed at showing different fields. The redundant images have been removed.

In 3E, it is unclear whether CXCR4 is expressed in PSCs or axons as there is expression in domains of overlapping green (SCs-GFP) and axons in red (Smi81, although what SMI81 stains is not defined). Maybe in situ hybridization is a better option to determine cell specific expression.

In situ hybridization of CXCR4 mRNA would not be useful in the present case, as it is likely that transcription of the receptor mRNA takes place in the motor neuron cell body in the spinal cord, not at the periphery.

We believe that the immunostaining of the receptor at the NMJ clearly shows its neuronal expression upon injury (please see the new representative images reported in panel C of Figure 3).

SMI81 refers to the monoclonal anti-SNAP 25 primary antibody used in the study. We apologize for the omission.

Minor points

Figure 1B panels labeled for Snap25 should be labeled as Vamp1.

Amended.

Figure 1 sample size is not recorded in the figure legend as in other figure.

Amended.

For figure 1A,B, the qualitative nature of the data is very striking, but quantification of presynaptic marker recovery would be useful to determine if variability exists and to what extent across neuromuscular junctions within the muscle.

Quantification of presynaptic markers recovery in LAL muscles exposed to α -LTx was reported in a recent publication of this lab (please refer to Duregotti et al., PNAS 2015).

Figure 1C may be kept but is somewhat unnecessary as an understanding of the experimental paradigm comes naturally from figure 1A,B.

Panel C has been removed from Figure 1, modified and reported as Figure EV1 in the revised manuscript.

In 3E, SMI81 should be Snap25 to be consistent with Figure 1.

Amended.

Referee #2 (Comments on Novelty/Model System):

I believe that the adopted model is fine. It would have been great to compare the poisoning done by treating the NMJ with alpha latrotoxin with another agent (e.g. taipoxin) acting via a completely different mechanism of action.

We agree with the Referee. Indeed, the two types of presynaptic neurotoxins act via a completely different biochemical mechanism of action. However, we and others have shown before that, eventually, the common toxic event that leads to nerve terminal degeneration is the entry of an excessive amount of Ca^{2+} . Therefore there is no reason to assume that the two types of neurotoxins will lead to a different regeneration program.

Referee #2 (Remarks):

This manuscript by Negro and colleagues describes an interesting transcriptomic approach aimed to the identification of novel signals involved in the functional recovery of the neuromuscular junction (NMJ) upon damage. The chosen insult is the treatment of the NMJ with alpha-latrotoxin, a presynaptic poison that elicits a long lasting inhibition of this specialized synaptic terminal. The authors found that stromal derived factor 1 (SDF-1 or CXCL12 alpha) is released following NMJ degeneration by schwann cells and promote regeneration via the activation of its receptor CXCR4. Sequestration of SDF-1 or inhibition of its receptor halts the functional recovery of the NMJ, suggesting that this signal transduction pathway is pivotal in NMJ regeneration. The results are presented in a clear, succinct manner and in logical order, which makes the manuscript easy to read. In general, the claims are supported by the figures provided. However, specific aspects of this manuscript should be revised prior to publication:

1. In contrast to the statement made in the main text and caption, both red channels in Figure 1A and B are labelled with SNAP25, rather than with SNAP25 and VAMP1. This should be amended.

We apologize for this mislabeling. The correct label is reported in the revised paper.

The specific treatment should be added to panel B.

Done

It would be useful to quantify the loss of the presynaptic markers using alpha-bungarotoxin as a mask. This analysis should address whether both plasma membrane and synaptic vesicle markers are depleted in the same extent.

In our recent publication (Duregotti et al., PNAS 2015) we measured the loss and subsequent reappearance of the presynaptic marker SNAP25 (mainly plasma membrane staining) as a mean to quantify motor axon terminal degeneration and regeneration upon intoxication. Duchen in the early '80s reported the complete and reversible fragmentation of motor axon terminals upon poisoning with the venom of the black widow spider. The net result is a presynaptic localized neurodegeneration, no matter whether plasma membrane and vesicles are destroyed to a different extent. Once the calcium influx occurs through the pores made by α -LTx in the plasma membrane, the complete degeneration of the nerve terminal (due to activation of degradative enzymes) takes place very rapidly.

2. The scheme in Figure 1C gives the wrong impression that samples are collected and/or imaged in a longitudinal rather than at fixed time points. This is misleading, especially for a scheme that should have the opposite goal. I suggest removing it.

The scheme has been removed from Figure 1 and a modified version is now presented as Figure EV1 of the revised manuscript.

3. No specific data emerging from the transcriptomic analysis have been provided (e.g. heat map) or if the raw data are going to be deposited in a suitable public database (e.g. Array Express at <http://www.ebi.ac.uk/arrayexpress>).

We are currently running an additional transcriptome analysis experimentally similar to the one that led to the identification of CXCL12 α as a possible hit, but performed with Illumina Technology, rather than Ion Torrent. Once completed, we will perform a broad bioinformatic analysis to identify activated and depressed pathways, and we will deliver the entire set of transcriptomic data to a public data base in the context of a novel submitted paper.

In the present study we singled out CXCL12 α hit (whose mRNA increases during neurodegeneration to return to basal levels when regeneration is almost complete) as a potential candidate in motor axon terminal regeneration given its role in the immune response and also in motor axon development. The validation of this hit with a specific antibody revealed that the molecule is indeed involved in NMJ recovery of function. It is validation by different means (we also performed CXCL12 α mRNA FISH at the NMJ following Referee 1 suggestion, reported in Fig1E of the revised manuscript) that makes one transcriptomic hit relevant and biologically significant. The remaining experiments described in the present paper were built on this validation.

4. The staining of CXCL12 alpha is poorly visible and should be represented in white in the central panels to increase the contrast. The authors should comment on CXCL12 alpha distribution since its localization is unclear: some puncta seem to be present within the schwann cells, whereas some others map in areas not labeled in green (top in the alpha-latrotoxin 4 h sample).

CXCL12 staining is now reported in white and a new confocal image is presented.

After a careful inspection of dozen of pictures we can safely affirm that CXCL12 puncta are inside Schwann cells only. The few spots outside Schwann cells in the former picture may be attributed to some aspecificity of the secondary antibody, and also to the basal autofluorescence of muscle tissue. In the revised manuscript we have reported different confocal images and the relative orthogonal projection that clearly show the localization of the chemokine in terminal Schwann cells. In addition, we have performed in situ hybridization of CXCL12 α mRNA at the NMJ (as suggested by Referee 1), that further confirms the expression of the chemokine mRNA in PSCs (Figure 1E).

Likewise the statement that CXCL12 alpha puncta are colocalized with Lysotracker should be tuned down, since this colocalization is partial at best (Figure EV1). Figure EV2 seems also redundant.

Given the CXCL12 α -positive spots, originally we wished to understand whether they were neutral classical secretory vesicles or acidic compartments destined to lysosomal exocytosis. This is the reason why we performed Lysotracker staining. Indeed, the chemokine appears to be accumulated in acidic structures, whose exact nature is currently under investigation. This point was arised also by Referee 1 and, accordingly, we have removed this image from the revised version. Indeed, establishing the nature of the secretory vesicle containing CXCL12 α within PSCs is beyond the scope of the present study, and will be tackled in the future.

5. In Figure 2A, are control samples treated with alpha-latrotoxin and either with saline or an equivalent dose of an irrelevant antibody? Please clarify the methodology in the main text.

The bar labelled as "Ctr" corresponds to EJPs of soleus muscles from mice locally injected with saline in the hind limb, whereas the "anti CXCL12 α " bar represents those from mice intraperitoneally injected with the sole anti-chemokine antibody. The methodology has been clarified accordingly in the revised manuscript (please see the Results section).

6. Figure 2D is not very informative, since the more informative quantification of these data is provided in panel E.

For clarity we have added a scheme of the microfluidic devices set-up and a new representative image (panel D of Figure 2).

Does treatment with CXCL12 alpha alter neuronal survival in the conditions described in this Figure?

We have performed viability assay on SCMNs exposed to 500 ng/ml rCXCL12 α for 5 DIV and found no differences in survival upon treatment.

Furthermore, is the effect on axonal elongation detected if CXCL12 alpha is only added in the somatic chamber?

When applied to the somatic chamber of microfluidic devices, rCXCL12 α failed to promote axonal elongation. Results of the proposed experiment are reported in Figure EV4 of the revised manuscript.

7. The resolution of Figure 3D is not optimal.

A new confocal image is reported in Figure 3A of the revised manuscript.

8. Does CXCR4 co-localize in motoneurons with classical growth cone markers?

Co-localization of CXCR4 with the growth cone marker GAP43 has been performed and a representative image is shown in Figure 3B of the revised manuscript.

Why is SMI81 used in Figure 3E instead of SNAP25?

We apologize. SMI81 identifies the anti-SNAP25 antibody employed in the present study. Label has now been changed accordingly.

2nd Editorial Decision

19 April 2017

Thank you for the submission of your revised manuscript to EMBO Molecular Medicine. We have now received the enclosed reports from the referees that were asked to re-assess it. As you will see the reviewers are now globally supportive and I am pleased to inform you that we will be able to accept your manuscript pending final editorial amendments [not detailed]:

Please submit your revised manuscript within two weeks. I look forward to seeing a revised form of your manuscript as soon as possible.

***** Reviewer's comments *****

Referee #1 (Remarks):

The authors have adequately addressed the reviewer's comments, which improved the overall quality and impact of the manuscript.

Referee #2 (Comments on Novelty/Model System):

as stated in the original review

Referee #2 (Remarks):

The authors have addressed all my comments in the revised manuscript, which is massively improved and now ready for publication.

2nd Revision - authors' response

03 May 2017

Authors made requested editorial changes.

Corresponding Author Name: Cesare Montecucco

Journal Submitted to: Embo Molecular Medicine

Manuscript Number: EMM-2016-07257